# Exploration of the Potential for Efficient Fiber Degradation by Intestinal Microorganisms in Diqing Tibetan Pigs

**Lijie Yang [1], Gang Wang [1], Junyan Zhou [1], Yuting Yang [1], Hongbin Pan [2], Xiangfang Zeng [1] and Shiyan Qiao [1,*]**

[1] State Key Laboratory of Animal Nutrition, College of Animal Science and Technology, China Agricultural University, Yuanmingyuan West Road 2, Haidian District, Beijing 100193, China; B20193040351@cau.edu.cn (L.Y.); B20173040313@cau.edu.cn (G.W.); BS20193040402@cau.edu.cn (J.Z.); B20213040355@cau.edu.cn (Y.Y.); zengxf@cau.edu.cn (X.Z.)

[2] Yunnan Provincial Key Laboratory of Animal Nutrition and Feed Science, Faculty of Animal Science and Technology, Yunnan Agricultural University, Kunming 650201, China; ynsdyz@ynau.edu.cn

\* Correspondence: qiaoshiyan@cau.edu.cn

**Abstract:** In order to study the potential for efficient fiber degradation by intestinal microorganisms in Diqing Tibetan pigs, we first investigated the dietary structure of Diqing Tibetan pigs in their original habitat, then 60 healthy adult Diqing Tibetan pigs were randomly divided into 2 groups with 6 replicates each and 5 pigs in each replicate. The content of neutral detergent fiber in treatment 1 and 2 were adjusted to 20% and 40%, respectively. The total tract digestibility of nutrients and the degradation efficiency of fecal microorganisms to different types of fiber were determined. Results showed that the composition and nutritional level of Diqing Tibetan pig original diet differed greatly in different seasons. The content of crude fiber in the original diet was as high as 12.3% and the neutral detergent fiber was 32.5% in April, while the content of crude fiber was 4.9% and the neutral detergent fiber was 13.3% in October. With the increase of dietary fiber level, the total tract apparent digestibility of dry matter, crude fiber, crude protein, acid detergent fiber, neutral detergent fiber, ether extract, and organic matter decreased significantly ($p < 0.05$), and the contents of acetic acid, propionic acid, butyric acid, isobutyric acid, valeric acid, and isovaleric acid in the feces were also significantly ($p < 0.05$) reduced. The ability of Diqing Tibetan pig fecal microorganisms to degrade neutral detergent fiber was significantly higher ($p < 0.05$) than "Duroc × Landrace × Yorkshire" pig. In addition, there was no significant difference ($p > 0.05$) in the degradation efficiency of the same type of fiber between NDF-20 and NDF-40 groups. Our results strongly suggested that Diqing Tibetan pigs have the potential to efficiently utilize fiber, and their unique intestinal microbial composition is the main reason for their efficient utilization of dietary fiber.

**Keywords:** Diqing Tibetan pigs; fiber compositions; fiber degradation; intestinal microorganisms; in vitro fermentation; non-starch polysaccharide



## 1. Introduction

Tibetan pigs are a rare plateau-type pig breed that have the capacity to adapt to the low oxygen environment of a plateau region with strong resistance to adversity. They are mainly found in high altitude areas (altitudes = 2500–3500 m) such as Yunnan and Tibet in China and are the pig breed with the highest altitude distribution in the world. At present, the research on the Diqing Tibetan pig mainly focuses on growth and development [1,2], meat quality [3], and reproductive performance [4,5], while relatively little research has been done on the mechanisms of their herbivory. It has been reported that forage could account for 90% of the dietary composition of Diqing Tibetan pigs under the condition of house feeding [6]. However, previous reports have mostly focused on the roughage tolerance of Diqing Tibetan pigs, while little has been reported on their ability to utilize fiber efficiently. Some researchers are attempting to screen microbial strains with fiber-degrading ability directly from gut microorganisms of Diqing Tibetan pigs [6,7]. The lack of basic data

poses great difficulties for subsequent investigations and many scientific hypotheses need to be reconfirmed.

Fibers, as the main component of cell walls, are the most widely distributed and abundant renewable resources on the Earth. Non-starch polysaccharides (NSPs), as the most important component of fiber besides lignin, play a vital role in the functional regulation of fiber. Pigs are unable to synthesize cellulase, hemicellulase, and pectinase, so the degradation of dietary fiber is largely dependent on fermentation and non-enzymatic hydrolysis by microorganisms [8]. Microorganisms in the pig's gut use dietary fiber as a primary carbon source and eventually convert it to SCFAs, which can provide 5–20% of the total energy requirement of the host [9–11]. However, fiber has long been defined as an anti-nutritional factor and high fiber levels in the diet will affect the digestibility of energy and protein and other nutrients and reduce feed conversion efficiency [12,13].

An appropriate level of fiber in the diet has a positive effect on promoting development of the animal intestine and its microecological balance [14–17]. Fiber can provide a good substrate for microorganisms in the hindgut of pigs [18,19], while the short chain fat acid (SCFA) produced by fermentation can defend against infections and pathogenic bacteria by lowering intestinal pH [20–22]. Fiber can play a role in metabolism, supply of energy, etc. and that role could be influenced by the degree of degradation by intestinal microbes. Therefore, the screening of strains with efficient fiber degradation is particularly important.

## 2. Materials and Methods

Management and design of the experiment followed animal care rules approved by the Department of China Agricultural University Animal Care and Use Ethics Committee.

### 2.1. Sample Collection of Original Diet

In order to carry out a preliminary investigation of the composition and nutritional level of the original diet of Diqing Tibetan pigs, we entered Diqing Tibetan Autonomous Prefecture (altitude = 3220 m) twice in April and October of the same year to collect samples of the original diet of Diqing Tibetan pigs and local feed ingredients. Samples were prepared as air-dried samples and brought back to the laboratory for chemical analysis.

### 2.2. Animals and Dietary Treatments

A total of 60 healthy adult Diqing Tibetan pigs of similar body condition (male, 10 months old) were randomly divided into 2 groups with 6 replicates each and 5 pigs in each replicate. Diqing Tibetan pig rations were formulated with reference to NY (2004) nutritional requirements, and the NDF levels of treatment 1 and 2 were adjusted to 20% and 40%, respectively (Table 1). The experimental period lasted for 6 days after the pre-experimental period of 10 days. Water and rations were provided ad libitum for piglets.

The external indicator method ($Cr_2O_3$) was used to evaluate the total tract apparent digestibility of nutrients in Diqing Tibetan pigs. Pigs were kept in a pig sty (10 pigs in each pig sty) and fed three times in the morning, noon, and evening every day. Due to the high level of NDF in our experimental diet, Diqing Tibetan pigs usually defecated quickly, within one hour after feeding. We assigned special personnel to be responsible for fecal collection in each pig sty to avoid microbial loss caused by prolonged exposure to the air.

The total quantity of feces excreted by each pig was collected at 7 a.m. every morning. Feces were weighed and mixed. Representative samples were sampled and stored at −20 °C. For crude protein analysis, feces excreted were preserved in 1:3 diluted sulfuric acid. Subsamples of feeds and feces were dried in a 65 °C oven until constant weight, then let the samples regain moisture for 72 hours under natural conditions. After that, the samples were finely ground by mortar and pestle to pass through a 1 mm screen and then stored in sealed containers for the analysis of digestibility.

**Table 1.** Composition and nutritional level of experimental diet (air-dry basis), %.

| Items | NDF | |
|---|---|---|
| | 20% | 40% |
| Ingredients | | |
| Corn | 37 | 8 |
| Soybean meal | 18.5 | 20 |
| Broad bean bran flakes | 12 | 11.5 |
| Silage tartary buckwheat | 7 | 0 |
| Highland barley powder | 11 | 5 |
| Highland barley hulls | 10.5 | 50 |
| Soybean oil | 0 | 2.5 |
| Limestone | 1.32 | 0.35 |
| NaCl | 0.14 | 0.14 |
| $CaHPO_4 \cdot 2H_2O$ | 1.35 | 1.51 |
| L-Lysine-HCl | 0.13 | 0 |
| DL-Methionine | 0.01 | 0 |
| L-Threonine | 0.03 | 0 |
| L-Tryptophan | 0.02 | 0 |
| Premix [1] | 1 | 1 |
| Total | 100 | 100 |
| Nutrient levels | | |
| Digestible energy, MJ/kg | 11.7 | 11.7 |
| CP | 14 | 13.9 |
| NDF | 20.1 | 40.2 |
| Ca | 1 | 1 |
| P | 0.5 | 0.5 |
| Lysine | 0.87 | 0.87 |
| Methionine + Cystine | 0.5 | 0.5 |
| Threonine | 0.6 | 0.6 |
| Tryptophan | 0.2 | 0.2 |

[1] The premix provided the following per kg of the diet: VA 6000 IU, $VD_3$ 1500 IU, VE 15 IU, $VK_3$ 1.5 mg, $VB_1$ 0.9 mg, $VB_2$ 3 mg, $VB_6$ 1.5 mg, $VB_{12}$ 10 μg, niacin 17 mg, pantothenic acid 9 mg, folic acid 0.32 mg, biotin 0.02 mg, choline chloride 350 mg, Fe 90 mg, Cu 8 mg, Mn 20 mg, Zn 50 mg, I 0.32 mg, Se 0.30 mg.

### 2.3. Chemical Analysis

According to AOAC international methods, crude protein (CP), dry matter (DM), ether extract (EE), ash, calcium (Ca), phosphorus (P), acid detergent fiber (ADF), and neutral detergent fiber (NDF) were determined using a fiber analyzer (Ankom-220, Ankom Products, New York, NY, USA). The chromium content was determined by wet digestion (nitric acid + perchloric acid, GB/T 13088-2006) using an atomic absorption spectrometer (Hitachi-Z-5000, Hitachi Group, Tokyo, Japan).

### 2.4. Analysis of Non-Starch Polysaccharides and SCFAs

Total non-starch polysaccharide (NSP), insoluble non-starch polysaccharide (INSP), soluble non-starch polysaccharide (SNSP), and their constituent sugars were determined by gas-liquid chromatography (GLC) for neutral sugars. Colorimetry was used to determine glucuronic acid (Bach, 1997). The GLC of constituent sugars was performed on an Agilent GC 6890 with a flow of 20 mL/min and split 40:1. A 30 m × 0.25 mm × 0.25 μm column (Agilent DB-225, film thickness 0.25 μm) was used. The temperature of column and detector were 220 °C and 250 °C, respectively.

The concentration of SCFA and lactate in the feces were analyzed using the method described in a previous report [23] and with slight modifications. About 0.5 g of fecal sample was put into a 10 mL centrifuge tube. Then, 8.0 mL of ultrapure water was added, mixed, and centrifuged at 4000 rpm to obtain the supernate where 160 μL supernatant was placed into a 10 mL centrifuge tube and fixed to 8 mL. The supernate was filtered using a 0.20 mm nylon membrane filter (Millipore, Bedford, OH, USA) and poured into a liquid

chromatography system (Agilent Technologies 1200, Agilent Technologies, Santa Clara, CA, USA).

### 2.5. Preparation of Fermentation

In order to avoid the short residence of digesta in the digestive tract caused by high fiber content and eliminate the influence of the foregut on the fiber digestibility, the in vitro fermentation tests of fecal microorganisms were carried out. Then, the role of microorganisms in fiber degradation was further verified by comparing the changes of NDF content in substrates before and after fermentation.

Exp.1. Feces from Diqing Tibetan pigs (the content of NDF in the feed are 20% and 40%, respectively) and Duroc × Landrace × Yorkshire pigs (DLY, BW = 55 kg, feed with commercial complete formula feed) were selected as starter culture. The total quantity of feces excreted by each pig was collected at 7 a.m. every morning. The samples needed to be quickly mixed under sterile conditions and stored in liquid nitrogen. In each treatment, the fecal samples of every 3 pigs were mixed into one for fermentation experiments. After treatment, fecal samples were mixed with PBS and glycerol and sealed in sterile sampling bags to form a sample system. Then, we filtered the system under anaerobic conditions to obtain bacterial liquid, which was used as the starter culture of fermentation test. The feed of Diqing Tibetan pigs and DLY pig were used as fermentation substrates and fermented anaerobically at 37 °C for 5 d. The system of sampling and fermentation are shown in Tables 2 and 3.

**Table 2.** The system of sampling and fermentation.

| Items | Sampling System | | | Fermentation System | | |
|---|---|---|---|---|---|---|
| Component | PBS | Sample | Glycerol | Bacterial fluid | Substrate | Culture medium [1] |
| Content | 150 mL | 30 g | 20 mL | 5 mL | 0.5 g | 82 mL |

[1] The medium consists of various microelements and vitamins for the growth of microorganisms. The detailed composition is shown in Table 3.

**Table 3.** Composition of culture medium in fermentation system.

| Items | Composition | Amount |
|---|---|---|
| $H_2O$ | $H_2O$ | 400 mL |
| Solution A, per 100 mL | $CaCl_2 \cdot 2H_2O$ 13.2 g, $MnCl_2 \cdot 4H_2O$ 10 g, $CoCl_2 \cdot 6H_2O$ 1 g, $8FeCl_3 \cdot 6H_2O$ 8 g | 0.1 mL |
| Solution B, per 1 L | $NH_4HCO_3$ 4 g, $NaHCO_3$ 35 g | 200 mL |
| Solution C, per 1 L | $Na_2HPO_4$ 5.7 g, $KH_2PO_4$ 6.2 g, $MgSO_4 \cdot 7H_2O$ 0.6 g | 200 mL |
| Resazurin, per 100 mL | 100 mg | 1 mL |
| Reducing agent, per 100 mL | NaOH 160 g, $Na_2S \cdot 9H_2O$ 625 mg | 40 mL |

Exp.2. Feces from Diqing Tibetan pigs (the content of NDF in the feed are 20% and 40%, respectively) were selected as starter culture. Wheat bran, oat bran, and highland barley hulls were used as fermentation substrates and fermented anaerobically at 37 °C for 5 d.

### 2.6. Detection of Bacteria by Crystal Violet Staining

Firstly, 10 uL bacterial liquid was dropped into the center of the clean slide and applied into a bacterial film with a sterile inoculation ring. Then, the side with bacterial film was placed upward and the water evaporated dry with a low fire. After the slide was cooled, an appropriate amount of ammonium oxalate crystal violet dye was added and incubated for 1 min. The dye solution was poured out and the slide washed with distilled water until there was no color of the dye solution in the flowing water. Finally, the excess water was removed with absorbent paper, observed, and recorded under the microscope. Sections of bacterial strains were observed at a magnification of 400× (Nikon ECLIPSE 80i, Tokyo, Japan). Bacterial strains were counted and classified as cocci, brevibacterium, and longbacterium.

*2.7. Statistical Analysis*

To determine differences among treatments, the data of digestive trial and fermentation trial were analyzed using *t*-tests procedures and one-way ANOVA procedures of GraphPad Prism 8.0.2, respectively. The data were initially analyzed using a completely randomized design. All statements of significance were based on the probability of $p < 0.05$.

**3. Results**

*3.1. Investigation on the Original Diet of Diqing Tibetan Pigs in Spring*

The results showed that the composition and nutritional level of Diqing Tibetan pigs' original diet differed greatly in different seasons. Nine kinds of ingredients, including wheat bran, corn, broad bean bran flakes, wheat bran flakes, potato, turnip, tartary buckwheat leaves, highland barley hulls, and highland barley powder, were used in April (Table 4). In the original diet, highland barley powder, tartary buckwheat leaves, turnip, and other local non-conventional feed ingredients accounted for 82.15%, with distinctive regional characteristics of the plateau. The results of nutritional analysis show that the content of CF in the original diet was 12.34% and NDF was 32.49%, which provides evidence for the ability of Diqing Tibetan pigs to tolerate roughage. However, whether Diqing Tibetan pigs have the ability to make efficient use of dietary fiber still needs to be further explored.

**Table 4.** Composition and nutrient level of original diet for Diqing Tibetan pigs (April, air-dry basis), %.

| Ingredients | Content | Nutrient Levels | Content |
|---|---|---|---|
| Wheat bran | 5.95 | DM | 91.59 |
| Corn | 7.14 | CP | 9.06 |
| Broad bean bran flakes | 4.76 | CA | 5.43 |
| Wheat bran flakes | 4.76 | EE | 1.95 |
| Potato | 10.71 | CF | 12.34 |
| Turnip | 15.48 | Ca | 1.07 |
| Tartary buckwheat leaves | 15.48 | ADF | 18.73 |
| Highland barley hulls | 5.95 | NDF | 32.49 |
| Highland barley powder | 29.76 | | |
| Total | 100 | | |

*3.2. Investigation on the Original Diet of Diqing Tibetan Pigs in Autumn*

The change of seasons led to a decrease in the amount of green vegetation and Diqing Tibetan pigs were changed from free-range to captive breeding. The dietary structure reflected this change in that it mainly consisting of six kinds of raw materials: corn, highland barley powder, silage tartary buckwheat, chicory, alfalfa, and broad bean bran flakes (Tables 5 and 6). At this time, the content of CF and NDF in the feed were 4.9% and 13.3%, respectively. Corn was added at 60% as the main source of energy in the diet, while chicory and alfalfa were added at 8% as the main source of protein in this diet.

*3.3. Selection of Candidate Fiber Sources for Digestive Trial*

The results showed that the total NSP content of the three candidate fiber sources was highland barley hulls (44.4%) > wheat bran (37.5%) > oat bran (20.4%); in terms of total INSP content, wheat bran (36.6%) > highland barley hulls (36.5%) > oat bran (16.3%), where the arabinose content was wheat bran (8.6%) > highland barley hulls (3.1%) > oat bran (2.4%), xylose content highland barley hulls (17.0%) > wheat bran (11.7%) > oat bran (3.2%), and glucose content highland barley hulls (11.7%) > wheat bran (7.7%) > oat bran (7.5%); and the total SNSP content, highland barley hulls (8.0%) > oat bran (4.1%) > wheat bran (0.9%) (Table 7). Due to the low SNSP content in the three candidate fiber sources, the compositional differences will not be described here.

**Table 5.** Nutrient composition of raw materials in the original diet of Diqing Tibetan pigs (October, air-dry basis), %.

| Items | Corn | Highland Barley Powder | Silage Tartary Buckwheat | Chicory | Alfalfa | Broad Bean Bran Flakes |
|---|---|---|---|---|---|---|
| DM | 88 | 89 | 96.1 | 93.9 | 95.8 | 91.3 |
| CP | 7.2 | 10.9 | 6.9 | 22.5 | 22.8 | 9 |
| CA | 1.1 | 1.7 | 17.7 | 14.3 | 12 | 8.9 |
| EE | 3.1 | 1.9 | 1.9 | 4.8 | 3.9 | 1.7 |
| CF | 3.3 | 3.8 | 24.6 | 13.5 | 19.1 | 30.9 |
| Ca | 1.7 | 1.8 | 3 | 3.1 | 3.7 | 3.2 |
| NDF | 18.6 | 46.7 | 45.8 | 33.2 | 30.4 | 42.5 |
| ADF | 3.7 | 6 | 29.9 | 30.3 | 20.7 | 32.9 |

**Table 6.** Composition and nutritional level of original diet for Diqing Tibetan pigs (October, air-dry basis), %.

| Ingredients | Content | Nutrient Levels | Content |
|---|---|---|---|
| Corn | 60 | DM | 92.2 |
| Highland barley powder | 8 | CP | 8.9 |
| Silage tartary buckwheat | 8 | EE | 3.4 |
| Chicory | 8 | CF | 4.9 |
| Alfalfa | 8 | NDF | 13.3 |
| Broad bean bran flakes | 8 | Ca | 2 |
| Total | 100 | P | 0.2 |

**Table 7.** Determination of fiber components in feed materials, %.

| Items | | Wheat Bran | Oat Bran | Highland Barley Hulls | Original Diet |
|---|---|---|---|---|---|
| INSP | | | | | |
| | Rhamnose | 6.555 | 2.105 | 2.955 | 2.697 |
| | Fucose | 0.118 | 0.126 | 0.174 | 0.159 |
| | Fructose | 0.015 | 0.009 | 0.015 | 0.021 |
| | Ribose | 0.372 | 0.045 | 0.159 | 0.139 |
| | Arabinose | 8.595 | 2.421 | 3.145 | 2.176 |
| | Xylose | 11.701 | 3.157 | 17.014 | 4.38 |
| | Mannose | 0.27 | 0.346 | 0.253 | 0.403 |
| | Galactose | 0.556 | 0.353 | 0.538 | 0.546 |
| | Glucose | 7.653 | 7.525 | 11.687 | 9.506 |
| | Uronic acid | 0.72 | 0.26 | 0.51 | 0.59 |
| | Total INSP | 36.554 | 16.346 | 36.451 | 20.619 |
| SNSP | | | | | |
| | Rhamnose | 0.127 | 0.148 | 1.234 | 1.311 |
| | Fucose halose | 0.02 | 0.05 | 0.24 | 0.287 |
| | Fructose | 0.001 | 0.002 | 0.044 | 0.042 |
| | Ribose | 0.024 | 0.004 | 0.046 | 0.039 |
| | Arabinose | 0.058 | 0.086 | 0.375 | 0.364 |
| | Xylose | 0.102 | 0.087 | 0.401 | 0.784 |
| | Mannose | 0.204 | 0.305 | 1.743 | 2.411 |
| | Galactose | 0.053 | 0.068 | 0.929 | 0.742 |
| | Glucose | 0.145 | 3.168 | 1.683 | 2.545 |
| | Uronic acid | 0.21 | 0.17 | 1.3 | 2.17 |
| | Total SNSP | 0.944 | 4.087 | 7.993 | 10.695 |
| Total NSP | | 37.498 | 20.433 | 44.444 | 31.314 |

*3.4. Total Tract Apparent Digestibility and SCFAs in Feces*

The results showed that the total tract apparent digestibility of DM, CF, CP, NDF, ADF, EE, and OM in Diqing Tibetan pigs decreased significantly ($p < 0.05$) with the increase of

dietary fiber level (Table 8). When the content of NDF was adjusted to 20% and 40%, the digestibility of NDF in Diqing Tibetan pigs was 49.64% and 40.13%, respectively. Table 9 shows that the contents of acetic acid, propionic acid, butyric acid, isobutyric acid, valeric acid, and isovaleric acid in the feces of Diqing Tibetan pigs were significantly ($p < 0.05$) reduced with the increase of fiber level in the diet.

**Table 8.** Effect of different dietary fiber levels on the digestibility of nutrients in Diqing Tibetan pigs, %.

| Items | NDF-20 | NDF-40 | SEM | *p*-Value |
|---|---|---|---|---|
| DM | 75.12 [a] | 67.36 [b] | 0.017 | <0.001 |
| CF | 46.22 [a] | 39.24 [b] | 0.017 | 0.006 |
| CP | 74.89 [a] | 69.85 [b] | 0.012 | 0.008 |
| NDF | 49.64 [a] | 40.13 [b] | 0.026 | 0.047 |
| ADF | 40.36 [a] | 23.83 [b] | 0.044 | 0.036 |
| EE | 49.92 [a] | 43.02 [b] | 0.016 | 0.003 |
| OM | 77.50 [a] | 68.47 [b] | 0.02 | <0.001 |

In the same row, values with different small letter superscripts mean significant difference ($p < 0.05$), while with no letter superscripts mean no significant difference ($p > 0.05$).

**Table 9.** Effect of different fiber levels on SCFAs in the feces of Diqing Tibetan pigs, mg/kg.

| Items | NDF-20 | NDF-40 | SEM | *p*-Value |
|---|---|---|---|---|
| Lactic acid | 23.94 | 23.67 | 1.34 | 0.932 |
| Acetic acid | 3171.42 [a] | 2372.86 [b] | 180.92 | <0.001 |
| Propionic acid | 1032.36 [a] | 776.73 [b] | 64 | 0.017 |
| Formic acid | 1.96 | 3.52 | 0.53 | 0.16 |
| Isobutyric acid | 76.67 [a] | 54.10 [b] | 6.05 | 0.039 |
| Butyric acid | 326.63 [a] | 231.80 [b] | 23.11 | 0.01 |
| Isovaleric acid | 34.29 [a] | 23.22 [b] | 2.82 | 0.021 |
| Valeric acid | 40.40 [a] | 25.63 [b] | 3.6 | 0.01 |
| Total | 4707.67 [a] | 3511.53 [b] | 270.55 | <0.001 |

In the same row, values with different small letter superscripts mean significant difference ($p < 0.05$), while with no letter superscripts mean no significant difference ($p > 0.05$).

*3.5. Fiber Degradation Capacity of Fecal Microorganisms*

The ability of Diqing Tibetan pig fecal microorganisms to degrade NDF in commercial full formula feed (NDF = 16.4%) and diet with 20% NDF content was significantly higher ($p < 0.05$) than DLY group, but there was no significant difference ($p > 0.05$) between NDF-20 and NDF-40 groups (Figure 1A,B). However, when the fermentation substrate was the Diqing Tibetan pig's diet with 40% NDF, the NDF degradation ability of NDF-40 group was higher ($p < 0.05$) than that of DLY and NDF-20 group, but there was no significant difference ($p > 0.05$) between DLY and NDF-20 group (Figure 1C).

The degradation ability of fecal microorganisms in Diqing Tibetan pigs was different for diverse types of fibers (Figure 2A). Among them, the degradation of NDF in oat bran was significantly higher ($p < 0.05$) than that of wheat bran and highland barley hulls in both NDF-20 and NDF-40 groups, while the degradation of NDF in wheat bran was significantly higher ($p < 0.05$) than that of highland barley hulls in both groups. When the substrates were wheat bran, oat bran, and highland barley hulls, neither NDF-20 nor NDF-40 group showed significant differences ($p > 0.05$) in the degradation rate of NDF (Figure 2B).

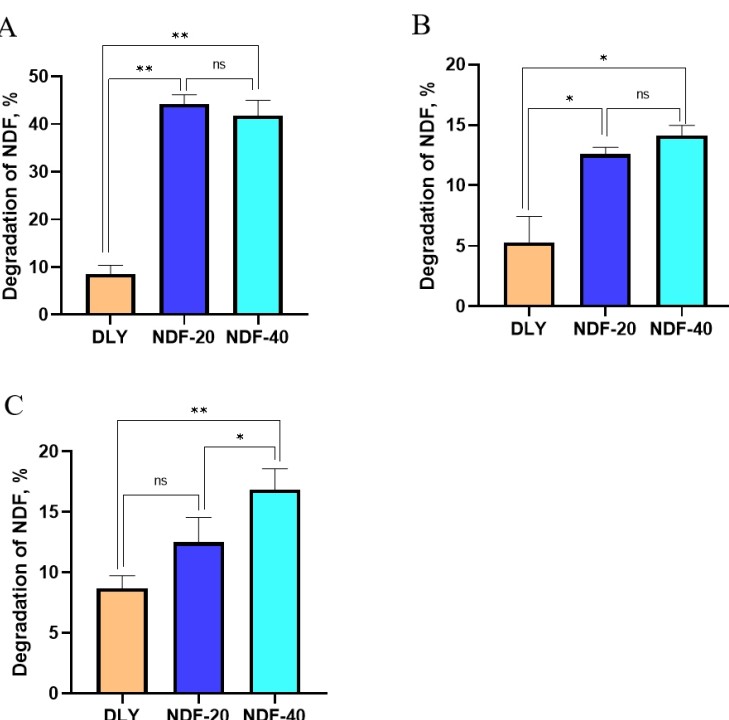

**Figure 1.** Comparison of NDF degradation ability between fecal microorganisms of Diqing Tibetan pigs and DLY pigs. (**A**) Commercial complete formula feeds of DLY pigs and Diqing Tibetan pig feed, of which NDF contents were (**B**) 20% and (**C**) 40%, were selected as fermentation substrates, and feces from Diqing Tibetan pigs and DLY were selected as starter culture. * $p < 0.05$; ** $p < 0.01$.

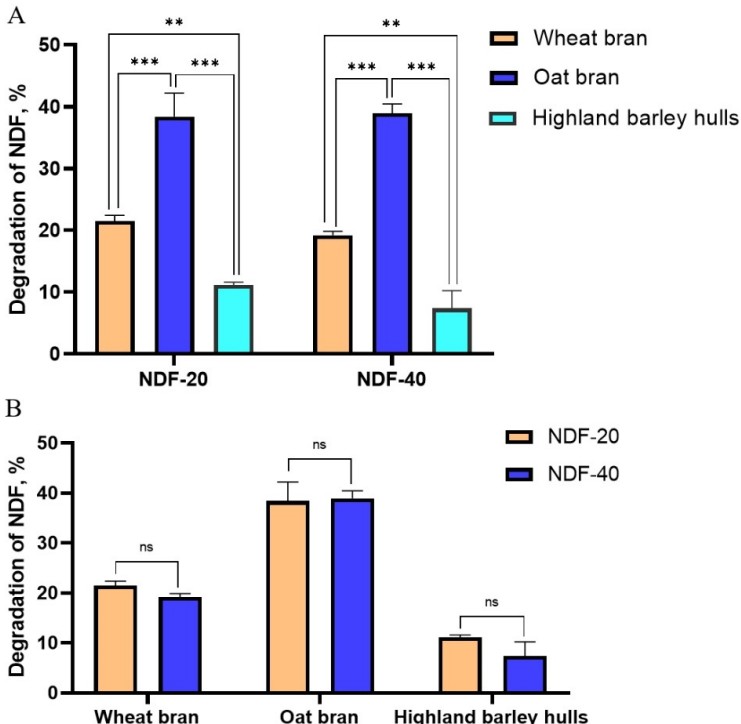

**Figure 2.** Comparison of degradation ability of Diqing Tibetan Pigs fecal microorganism to different types of fiber. (**A**,**B**) Wheat bran, oat bran, and highland barley hulls were selected as fermentation substrates, and feces from Diqing Tibetan pigs (the content of NDF in the feed are 20% and 40%, respectively) were selected as starter culture. ** $p < 0.01$, and *** $p < 0.005$.

In addition, we observed the fecal microorganisms of DLY and Diqing Tibetan pigs by microscopy, and we found that the number of live bacteria was NDF-40, NDF-20, and DLY in descending order, and the microscopic morphology of fecal microorganisms of DLY were different from that of Diqing Tibetan pigs (Figure 3). The fecal microorganisms of DLY pigs were mainly in the form of short rods and single morphology, while the fecal microorganisms of Diqing Tibetan pigs were composed of cocci, short rods, and long rods, and the number was higher. Compared with NDF-20, the morphology of NDF-40 was similar, but the number of NDF-40 groups was more. It is suggested that the specific flora structure and quantity of manure microorganisms of Diqing Tibetan pigs is one of the important factors to ensure its high efficiency fiber degradation ability.

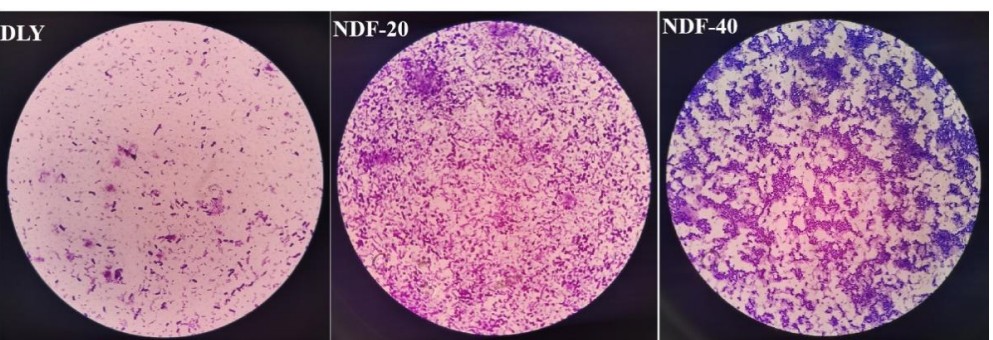

**Figure 3.** Microscopic observation (400×) on fecal microorganisms of DLY and Diqing Tibetan pigs by crystal violet staining.

## 4. Discussion

### 4.1. Investigation of Original Diet and Determination of Experimental Diet

As a typical plateau-type pig breed, Diqing Tibetan pigs have gradually acquired excellent characteristics such as rough feeding resistance, disease resistance, and cold tolerance under the long-term natural selection conditions of oxygen deprivation, cold, and grazing [24–26]. Research on fiber degradation ability of Diqing Tibetan pig has become popular. Many researchers believe that although Diqing Tibetan pigs have the ability to tolerate rough feeding, they cannot make good use of dietary fiber. In addition, altitude, temperature, oxygen concentration, and diet are important reasons for the inconsistency of research results. Therefore, it is necessary for us to return to the origin of the proposition and verify whether Diqing Tibetan pigs have efficient dietary fiber degradation ability. The investigation of habitat environment and dietary structure of Diqing Tibetan pigs is particularly important. The understanding of Diqing Tibetan pig species resources is of great practical significance for improving the living standards of ethnic minorities, achieving the revitalization of pastoral areas, and the stability of the frontier.

We entered Diqing Tibetan Autonomous Prefecture twice in April and October of the same year to investigate the diets of Diqing Tibetan pigs. The results showed that the original diet composition of Diqing Tibetan pigs had distinct seasonal dependence and varied greatly in nutritional composition. Furthermore, the NDF level in the diet of Diqing Tibetan pigs in April was much higher than that of the DLY complete formula feed at the same growth stage [27], while the NDF level in October was close to that of the DLY pigs, which was closely related to the local climate and grazing conditions. Therefore, it is reasonable to put forward the scientific hypothesis that Diqing Tibetan pigs may possess a strong roughage tolerance.

The investigation of original diet provides a baseline for the formulation of experimental diets for animals. However, different sources of fiber under the same fiber level conditions have different effects on the digestibility of the animals [28,29]. In addition, studies have shown that higher levels of INSP accelerate the passage of digesta through the digestive tract, while higher levels of SNSP will adhere to the digesta surface and form a nutritional barrier, both of which are detrimental to the utilization of nutrients [30–32]. It

has also been shown that NSP can act as a prebiotic to maintain the balance and homeostasis of the intestinal flora and can act as a targeting regulator to reshape the flora structure. NSP is thus a double-edged sword, the composition of its components is the key to its pros and cons. Therefore, wheat bran, oat bran, and highland barley hulls were selected as candidate fiber sources and their fiber components were determined. Under the conditions of this experiment, the NSP level in the original diet of Diqing Tibetan pigs in April was 31.3%, while the NSP contents of wheat bran, oat bran, and highland barley hulls were 37.5%, 20.4%, and 44.4%, respectively. Therefore, in order to meet the design requirements of 20% and 40% NSP content in the experimental diets, to avoid large differences in the proportion of raw materials added due to fiber level adjustment, and to mimic the original Tibetan pig diet to the greatest extent, we proposed to select highland barley hulls as the main fiber source for the experimental diets.

### 4.2. Total Intestinal Apparent Digestibility and SCFAs in Feces

The utilization of fiber by pigs mainly depends on the fermentation and non-enzymatic hydrolysis of their intestinal microorganisms [8]. Studies have shown that the number of fibrinolytic bacteria in the intestine of pigs gradually increases with increasing age, and the number of fibrinolytic bacteria in the intestine of growing pigs is about $10^8$ CFU/g, while that of adult sows can reach six times that of growing pigs [33]. Studies on pigs showed that, at 2–3 weeks post-weaning, the rapidly developing microbiotas appeared to reach a developmental milestone as a relative degree of stability was evident [34]. Therefore, considering that the weaning time of Diqing Tibetan piglets is 3 months old, it is reasonable to choose Diqing Tibetan pigs around 10 months old for the study.

Fiber is only partially digested in the pig and is less digestible than DM, CP, and EE. A portion of the fiber can be digested before reaching the end of the ileum, while the majority is degraded by microorganisms in the latter part of the digestive tract [35]. The results of this study are slightly lower than the total tract apparent digestibility of NDF (52.9%) when wheat bran was used as the fiber source (NDF = 14.7%) in our previous study [28]. In order to simulate the original diet of Diqing Tibetan pigs to the greatest extent possible, the raw materials used in the experiment, such as highland barley hulls, broad bean bran flakes, and silage tartary buckwheat, were fed after being cut short (2–3 cm in length). Therefore, it is reasonable to presume that the larger feed size was an important factor affecting the digestibility of nutrients in Diqing Tibetan pigs. The abundant SCFAs in the intestine mainly came from the fermentation of fiber by fiber-degrading bacteria [36,37]. We also found that acetic acid, propionic acid, butyric acid, isobutyric acid, valeric acid, and isovaleric acid in the feces of Diqing Tibetan pigs were reduced with the increase of fiber content in the ration. Among them, butyric acid is mainly absorbed by colonic epithelial cells, while propionic acid and valeric acid are absorbed into the blood and then enter the liver for the propionic acid sugar isomerization process, and acetic acid is utilized by muscle and adipose tissue [38,39]. Although dietary fiber can theoretically provide a richer substrate for the growth of intestinal microorganisms in Diqing Tibetan pigs, too much INSP will greatly shorten the time for the passage of digesta through the intestine and reduce the time for sufficient contact between the flora and nutrients such as dietary fiber. Moreover, too much SNSP can also form a nutritional barrier by adhering to the digesta surface, which in turn affects the utilization of nutrients by the organism. It is suggested that the difference of fiber content is one of the important factors affecting the digestibility of nutrients such as NDF in Diqing Tibetan pigs.

### 4.3. Comparison of Fiber Degradation Ability of Fecal Microorganisms

Our group found that the abundance of *Ruminococcaceae*, a fiber-digesting bacteria in the colon of Rongchang sows during gestation, lactation, and the nulliparous period, was higher than that of Landrace, indicating that the intestinal microorganisms of local pig breeds may have great potential for efficient utilization of dietary fiber [40]. The efficient utilization of fiber by Diqing Tibetan pigs is closely related to their intestinal

microorganisms, but the mechanism is not yet clear [41,42]. The in vitro fermentation test with fecal microorganisms avoids the short residence time of digesta in the intestine due to high INSP content in the feed and the reduced digestibility due to excessive particle size, while eliminating interference of the anterior alimentary canal to the diet. Studies have shown that dietary changes play an important role in remodeling the structure of the intestinal flora [43–45], and that the intestinal flora is able to adapt to new dietary habits of the host by changing its structure [46,47]. The in vitro degradation of NDF in different types of diets by fecal microorganisms was higher in Diqing Tibetan pigs than in DLY pigs under the conditions of our experiment. Furthermore, fecal microorganisms from different sources have a relatively high ability to degrade NDF in the host test diet, which may be due to the different types of diet prompting the change in the structure of the host intestinal flora toward a more adapted diet.

The coefficient of variation in the digestibility of fiber from different sources is large in pigs [48], and the composition of fiber directly affects its physicochemical properties, which in turn affects digestibility [49]. In our previous study, we analyzed the fiber fractions of wheat bran, soybean bark, oat bran, palm kernel residue, ramie, fermented ramie, and bamboo meal and found that diets with the same fiber content but different fiber fractions had different effects on growth performance, digestibility, and fecal microorganisms of weaned piglets [28]. Under the present experimental conditions, the degradation ability of manure microorganisms of Diqing Tibetan pigs was different for fibers from different sources, among which oat bran was the highest, which might be related to the higher content of SNSP in oat bran. In contrast, the higher content of INSP in highland barley hulls and wheat bran may be one of the important factors for their lower degradation rate. Compared with SNSP, the components of INSP possessed a tighter connection, in which cellulose was connected by intermolecular hydrogen bonds into a supramolecular structure that was difficult to be broken by a variety of cellulases, thus making it more difficult to be degraded [50–53]. Furthermore, we found that the degradation ability of fecal microorganisms of Diqing Tibetan pigs to the same fiber source did not show statistical differences when the fiber content in diet was changed. We speculate that this is due to the fact that diets with high and low fiber levels changed the abundance of the intestinal flora, while the species did not change, resulting in the re-domestication of the flora structure during the in vitro fermentation. Therefore, combined with the previous results, we can speculate that the differences in nutrient digestibility in Diqing Tibetan pigs when fed diets with different fiber levels are mainly due to the nutrient barrier of SNSP and the accelerated circulation of digesta by INSP.

## 5. Conclusions

Our results strongly suggested that Diqing Tibetan pigs have the potential to efficiently utilize fiber, and their unique intestinal microbial composition is the main reason for their efficient utilization of dietary fiber. The results of the study provide a theoretical basis for an in-depth understanding of the biological characteristics of Diqing Tibetan pigs and fully exploit the resource advantages of its intestinal microorganisms. Furthermore, our results are of great significance to improve the utilization efficiency of fiber feed, save renewable resources, and alleviate the situation of human-animal food competition. However, the composition of key gut microorganisms for efficient fiber utilization in Diqing Tibetan pigs and the contribution and mechanism of these microorganisms in the degradation of fiber fraction need further study.

**Author Contributions:** Conceptualization, S.Q. and X.Z.; methodology, X.Z.; software, L.Y. and G.W.; validation, L.Y. and J.Z.; formal analysis, L.Y.; investigation, Y.Y.; resources, S.Q.; data curation, L.Y.; writing—original draft preparation, L.Y.; writing—review and editing, X.Z.; visualization, H.P.; supervision, S.Q.; project administration, S.Q.; funding acquisition, S.Q. All authors have read and agreed to the published version of the manuscript.

**Funding:** National Natural Science Foundation of China (U180220167).

**Institutional Review Board Statement:** The study was conducted according to the guidelines of the Declaration of Helsinki, and approved by the Ethics Committee of the Department of China Agricultural University Animal Care and Use Ethics Committee (AW12401202-1-1; 2020.04.30).

**Informed Consent Statement:** Not applicable.

**Data Availability Statement:** Not applicable.

**Acknowledgments:** This work was supported by the National Natural Science Foundation of China (U180220167) and the Beijing Swine Innovation Team of Modern Agriculture Industry Technological System.

**Conflicts of Interest:** The authors declare that they have no competing interests.

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
