# Peer review of "Exploration of the Potential for Efficient Fiber Degradation by Intestinal Microorganisms in Diqing Tibetan Pigs"

_fermentation, doi:10.3390/fermentation7040275_

Round 1
Reviewer 1 Report
In the research article “Research on an ancient species in the highlands: exploration of the potential for efficient fiber degradation by intestinal microorganisms in Diqing Tibetan pigs”, authors have revealed that Diqing Tibetan pigs have the potential to efficiently utilize fiber, and their unique intestinal microbial composition is the main reason for their efficient utilization of dietary fiber.
The title of the manuscript suits the journal.
The abstract presents a precise summary of the research findings in a clear way.
The keywords are appropriate yet should be in alphabetical order.
The language used in the manuscript is understandable.
The suitable sizes of samples have been used during experiments.
The experimental methods and study design are pertinent to rejoinder the study queries. The methodology is comprehensively discussed and research may be repeated.
The statistical analysis used is apposite and correctly elaborated.
The figures and tables represent current results.
The formerly executed research work has been discussed and current findings have been equated.
The appropriate citations have been used to support the statements.
The findings of the study support the conclusions.
The manuscript may be a suitable candidate for publication.
Author Response
Reply to Editor and reviewers
On behalf of my co-authors, we sincerely thank you for giving us an opportunity to revise our manuscript, and deeply appreciate editor and reviewers for their positive and constructive comments and suggestions on our manuscript entitled “Research on an ancient species in the highlands: exploration of the potential for efficient fiber degradation by intestinal microorganisms in Diqing Tibetan pigs”.
We have checked the revised manuscript carefully, and look forward to hear from you soon. If you have any questions, please don’t hesitate to let us know. Appended to this letter is our point-by-point response to the comments raised by the reviewers. The comments are reproduced and our responses are given directly afterward in red color.
Reviewer #1: In the research article “Research on an ancient species in the highlands: exploration of the potential for efficient fiber degradation by intestinal microorganisms in Diqing Tibetan pigs”, authors have revealed that Diqing Tibetan pigs have the potential to efficiently utilize fiber, and their unique intestinal microbial composition is the main reason for their efficient utilization of dietary fiber.
The title of the manuscript suits the journal.
The abstract presents a precise summary of the research findings in a clear way.
The keywords are appropriate yet should be in alphabetical order.
- We are grateful for the suggestion. The keywords have been arranged in alphabetical order.
The language used in the manuscript is understandable.
The suitable sizes of samples have been used during experiments.
The experimental methods and study design are pertinent to rejoinder the study queries. The methodology is comprehensively discussed and research may be repeated.
The statistical analysis used is apposite and correctly elaborated.
The figures and tables represent current results.
The formerly executed research work has been discussed and current findings have been equated.
The appropriate citations have been used to support the statements.
The findings of the study support the conclusions.
The manuscript may be a suitable candidate for publication.
- Thanks for the reviewer’s comments. We have checked the revised manuscript carefully, and look forward to hear from you soon. If you have any questions, please don’t hesitate to let us know.

Reviewer 2 Report
Some comments and questions about the manuscript:
The title, perhaps better less literary, without “Research on an Ancient Species in the Highlands”?
In section 2.2. Animals and Dietary Treatments, 60 pigs of the test breed are given, there is no information about DLY breed mentioned in section 2.5.
Apart from the statement that they were adult animals in a similar condition, maybe it is worth providing information about their age and sex? When and how the fecal samples were collected? It is worth describing fecal sampling in more detail and precisely.
Part b of graph 2 is rather unnecessary, the text can provide information about the lack of significant differences in the degradation of NDF, if its level differs depending on its source.
The results are discussed in detail and properly interpreted, the illustrative material is clear, and the conclusions are correctly constructed.
The presented work should be assessed positively. Taking into account the cognitive and scientific value, it can be concluded that the paper may be published in the Fermentation journal, after possible minor corrections resulting from the comments presented.
Author Response
Reviewer #2: Some comments and questions about the manuscript:
On behalf of my co-authors, we sincerely thank you for giving us an opportunity to revise our manuscript, and deeply appreciate you for the positive and constructive comments and suggestions on our manuscript. We have checked the revised manuscript carefully, and look forward to hear from you soon. If you have any questions, please don’t hesitate to let us know.
The title, perhaps better less literary, without “Research on an Ancient Species in the Highlands”?
- Thanks for the kind remind. We have revised the title of the manuscript as suggested by the reviewer.
In section 2.2. Animals and Dietary Treatments, 60 pigs of the test breed are given, there is no information about DLY breed mentioned in section 2.5.
- Thanks for the editor’s comments. In the section of 2.5, we conducted an in vitro fermentation experiment of fecal microorganisms. The DLY mentioned in our experiment was Duroc×Landrace ×Yorkshire ternary hybrid pig, and is also the largest number of commercial pigs in the world. In our experimental, in order to evaluate the difference between Diqing Tibetan pigs and DLY under normal feeding conditions, we did not carry out special treatment on DLY, but simply collected feces from pigs fed commercial feed. According to the reviewer's suggestions, the necessary information has been supplemented in the manuscript.
Apart from the statement that they were adult animals in a similar condition, maybe it is worth providing information about their age and sex? When and how the fecal samples were collected? It is worth describing fecal sampling in more detail and precisely.
- We thank the reviewer for the kind remind. According to the reviewer's suggestions, “Total quantity of feces excreted by each pig was collected at seven every morning. Feces were weighed and mixed. Representative samples were sampled and stored at -20 ℃. For crude protein analysis, feces excreted were preserved in 1:3 diluted sulfuric acid. Subsamples of feeds and feces were dried in a 65 ℃ oven until constant weight, then let the samples regain moisture for 72 hours under natural conditions. After that, the samples were finely ground by mortar and pestle to pass through a 1 mm screen and then stored in sealed containers for the analysis of digestibility.” were added in the section of 2.2. Also, “Total quantity of feces excreted by each pig was collected at seven every morning.The samples should be quickly mixed under sterile conditions and stored in liquid nitrogen.” were added in the section of 2.5.
Part b of graph 2 is rather unnecessary, the text can provide information about the lack of significant differences in the degradation of NDF, if its level differs depending on its source.
- Thanks for the kind remind. In the part B of the figure 2, we just want to conclud the fact that diets with high and low fiber levels changed the abundance of the intestinal flora, while the species did not change, resulting in the re-domestication of the flora structure during the in vitro fermentation. Theoretically, high fiber diet may enrich more strains with fiber degradation ability, but this result was not shown in in vitro fermentation test. What’s more, the difference of fiber sources is the difference of fiber components, which leads to the difference of microbial community, which is our ongoing research content.
The results are discussed in detail and properly interpreted, the illustrative material is clear, and the conclusions are correctly constructed.
The presented work should be assessed positively. Taking into account the cognitive and scientific value, it can be concluded that the paper may be published in the Fermentation journal, after possible minor corrections resulting from the comments presented.
